# NeRN - Learning Neural Representations for Neural Networks

**Maor Ashkenazi**[1]*, **Zohar Rimon**[2], **Ron Vainshtein**[2], **Shir Levi**[3], **Elad Richardson**[3],
**Pinchas Mintz**[3], **Eran Treister**[1]
[1]Ben-Gurion University of the Negev  [2]Technion - Israel Institute of Technology  [3]Penta-AI

## Abstract

Neural Representations have recently been shown to effectively reconstruct a wide range of signals from 3D meshes and shapes to images and videos. We show that, when adapted correctly, neural representations can be used to directly represent the weights of a pre-trained convolutional neural network, resulting in a Neural Representation for Neural Networks (NeRN). Inspired by coordinate inputs of previous neural representation methods, we assign a coordinate to each convolutional kernel in our network based on its position in the architecture, and optimize a predictor network to map coordinates to their corresponding weights. Similarly to the spatial smoothness of visual scenes, we show that incorporating a smoothness constraint over the original network's weights aids NeRN towards a better reconstruction. In addition, since slight perturbations in pre-trained model weights can result in a considerable accuracy loss, we employ techniques from the field of knowledge distillation to stabilize the learning process. We demonstrate the effectiveness of NeRN in reconstructing widely used architectures on CIFAR-10, CIFAR-100, and ImageNet. Finally, we present two applications using NeRN, demonstrating the capabilities of the learned representations.

## 1 Introduction

In the last decade, neural networks have proven to be very effective at learning representations over a wide variety of domains. Recently, NeRF (Mildenhall et al., 2020) demonstrated that a relatively simple neural network can directly learn to represent a 3D scene. This is done using the general method for neural representations, where the task is modeled as a prediction problem from some coordinate system to an output that represents the scene. Once trained, the scene is encoded in the weights of the neural network and thus novel views can be rendered for previously unobserved coordinates. NeRFs outperformed previous view synthesis methods, but more importantly, offered a new view on scene representation. Following the success of NeRF, there have been various attempts to learn neural representations on other domains as well. In SIREN (Sitzmann et al., 2020) it is shown that neural representations can successfully model images when adapted to handle high frequencies. NeRV (Chen et al., 2021) utilizes neural representations for video encoding, where the video is represented as a mapping from a timestamp to the pixel values of that specific frame.

In this paper, we explore the idea of learning neural representations for *pre-trained* neural networks. We consider representing a Convolutional Neural Network (CNN) using a separate predictor neural network, resulting in a neural representation for neural networks, or NeRN. We model this task as a problem of mapping each weight's coordinates to its corresponding values in the original network. Specifically, our coordinate system is defined as a (Layer, Filter, Channel) tuple, denoted by $(l, f, c)$, where each coordinate corresponds to the weights of a $k \times k$ convolutional kernel. NeRN is trained to map each input's coordinate back to the original kernel weights. One can then reconstruct the original network by querying NeRN over all possible coordinates.

While a larger predictor network can trivially learn to overfit a smaller original network, we show that successfully creating a compact implicit representation is not trivial. To achieve this, we propose methods for introducing smoothness over the learned signal, i.e. the original network weights,

---

*Email: maorash@post.bgu.ac.il. Code avaliable at: `https://github.com/maorash/NeRN`.

either by applying a regularization term in the original network training or by applying post-training permutations over the original network weights. In addition, we design a training scheme inspired by knowledge distillation methods that allows for a better and more stable optimization process.

Similarly to other neural representations, a trained NeRN represents the weights of the specific neural network it was trained on, which to the best of our knowledge differs from previous weight prediction papers such as Ha et al. (2016); Schürholt et al. (2021); Knyazev et al. (2021); Schürholt et al. (2022). We demonstrate NeRN's reconstruction results on several classification benchmarks. Successfully learning a NeRN provides some additional interesting insights. For example, a NeRN with limited capacity must prioritize the original weights during training. This can then be explored, using the reconstruction error, to study importance of different weights. In addition to our proposed method and extensive experiments, we provide a scalable framework for NeRN built using PyTorch (Paszke et al., 2019) that can be extended to support new models and tasks. We hope that our proposed NeRN will give a new perspective on neural networks for future research.

## 2 RELATED WORK

**Neural representations** have recently proven to be a powerful tool in representing various signals using coordinate inputs fed into an MLP (multilayer perceptron). The superiority of implicit 3D shape neural representations (Sitzmann et al., 2019; Jiang et al., 2020; Peng et al., 2020; Chabra et al., 2020; Mildenhall et al., 2020) over previous representations such as grids or meshes has been demonstrated in Park et al. (2019); Chen & Zhang (2019); Genova et al. (2020). Following NeRF's success, additional applications rose for neural representations such as image compression (Dupont et al., 2021), video encoding (Chen et al., 2021), camera pose estimation (Yen-Chen et al., 2021) and more. Some of these redesigned the predictor network to complement the learned signal. For example, Chen et al. (2021) adopted a CNN for frame prediction. In our work, we adopt a simple MLP while incorporating additional methods to fit the characteristics of convolutional weights.

**Weight prediction** refers to generating a neural network's weights using an additional predictor network. In Ha et al. (2016) the weights of a larger network are predicted using a smaller internal network, denoted as a HyperNetwork. The HyperNetwork is trained to directly solve the task, while also learning the input vectors for parameter prediction. Deutsch (2018) followed this idea by exploring the trade-off between accuracy and diversity in parameter prediction. In contrast, we aim to directly represent a *pre-trained* neural network, using fixed inputs. Several works have explored the idea of using a model dataset for weight prediction. For instance, Schürholt et al. (2021) proposes a representation learning approach for predicting hyperparameters and downstream performance. Schürholt et al. (2022) explored a similar idea for weight initialization while promoting diversity. Zhang et al. (2018a); Knyazev et al. (2021) leverage a GNN (graph neural network) to predict the parameters of a previously unseen architecture by modeling it as a graph input.

**Knowledge distillation** is mostly used for improving the performance of a compressed network, given a pre-trained larger teacher network. There are two main types of knowledge used in student-teacher learning. First, response-based methods (Ba & Caruana, 2014; Hinton et al., 2014; Chen et al., 2017; 2019) focus on the output classification logits. Second, feature-based methods (Romero et al., 2015; Zagoruyko & Komodakis, 2017) focus on feature maps (activations) throughout the network. The distillation scheme can be generally categorized as offline (Zagoruyko & Komodakis, 2017; Huang & Wang, 2017; Passalis & Tefas, 2018; Heo et al., 2019; Mirzadeh et al., 2020; Li et al., 2020) or online (Zhang et al., 2018b; Chen et al., 2020; Xie et al., 2019). In our work, we leverage offline response and feature-based knowledge distillation for guiding the learning process.

## 3 METHOD

In this work we focus on representing convolutional classification networks. Our overall pipeline is presented in Figure 1, with extended details below on the design choices and training of NeRN.

### 3.1 DESIGNING NeRNs

Similar to other neural representations, at its core, NeRN is composed of a simple neural network, whose input is some positional embedding representing a weight coordinate in the original network

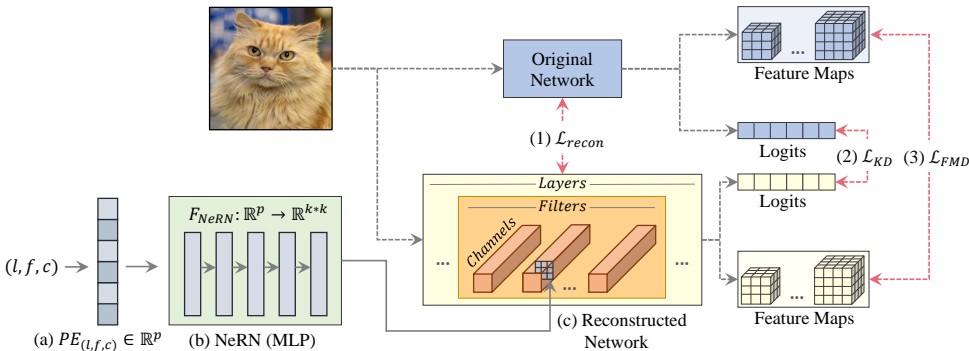

Figure 1: The NeRN pipeline. Input coordinates are transformed to positional embeddings **(a)**, which are mapped through NeRN **(b)** to obtain the predicted weights of a corresponding convolutional kernel. The predicted weights compose the reconstructed network **(c)**. NeRN is trained using a combination of three losses; **(1)** reconstruction loss between the original and reconstructed weights, **(2)** knowledge distillation loss between the original and predicted classification logits, and **(3)** feature map reconstruction loss between the original and predicted feature maps.

and whose output is the weight values at that coordinate. The predicted weights on all possible coordinates compose the predicted neural network, denoted as the reconstructed network.

**I/O modeling**  We propose to learn a mapping between a 3-tuple $(l, f, c)$ to the $k \times k$ kernel at channel $c$ of filter $f$ in layer $l$. Since the output size of NeRN is fixed, we set it to the largest kernel size in the original network, and sample from the middle when predicting smaller kernels. We model convolutional layers only, and not others such as fully-connected or normalization layers, as their parameters are of negligible size compared to the convolution weights (see Appendix C).

**Positional embeddings**  Similarly to preceding neural representation works (Nguyen-Phuoc et al., 2022; Chen et al., 2021; Tancik et al., 2020), the input coordinates are first mapped to a high dimensional vector space. By using a high dimensional space, NeRN is able to represent high-frequency variations in the learned signal. We adopt the positional embeddings used in Vaswani et al. (2017),

$$PE_{(l,f,c)} = concat\left(\gamma\left(l\right), \gamma\left(f\right), \gamma\left(c\right)\right), \tag{1}$$
$$\gamma\left(v\right) = \left[\sin(b^0 \pi v), \cos(b^0 \pi v), \dots, \sin(b^{N-1} \pi v), \cos(b^{N-1} \pi v)\right]$$

where $b$, and $N$ represent the base frequency and the number of sampled frequencies, respectively.

**Architecture**  The NeRN predictor is a 5-layer MLP, which is a simplified version of the architecture used in Park et al. (2019). We omit the internal concatenation of the positional embedding, as it did not change our empirical results. The hidden layer size is fixed throughout the network.

### 3.2  TRAINING NERNs

In order to train a NeRN we need to define a set of loss functions. The most basic loss is a reconstruction loss between the original and reconstructed network's weights. However, it is clear that some weights have more effect on the network's activations and output. Hence, we introduce two additional losses: a Knowledge Distillation (KD) loss and a Feature Map Distillation (FMD) loss. As presented in subsection 4.6, the reconstruction loss alone yields respectable accuracy, and the additional losses improves on it, promotes faster convergence and stabilizes the training process. Notice that our model is trained with no direct task loss and thus does not need access to labeled data. We further show in section 4.4 that NeRN might not require any data at all.

The objective function for training NeRNs is comprised of the following:

$$\mathcal{L}_{objective} = \mathcal{L}_{recon} + \alpha \mathcal{L}_{KD} + \beta \mathcal{L}_{FMD}, \tag{2}$$

where $\mathcal{L}_{\mathrm{recon}}$, $\mathcal{L}_{KD}$, and $\mathcal{L}_{FMD}$ denote the reconstruction, knowledge distillation and feature map distillation losses, respectively. The $\alpha$ and $\beta$ coefficients can be used to balance the different losses.

The weight reconstruction loss, $\mathcal{L}_{recon}$, is defined as:

$$\mathcal{L}_{recon} = \frac{1}{|\mathbf{W}|} \|\mathbf{W} - \hat{\mathbf{W}}\|_2, \tag{3}$$

where $\mathbf{W} = \left[ \mathbf{w}^{(0)} \mathbf{w}^{(1)} \ldots \mathbf{w}^{(L)} \right]$ and $\mathbf{w}^{(l)}$ is the tensor of layer $l$'s convolutional weights in the original network. Similarly, $\hat{\mathbf{W}} = \left[ \hat{\mathbf{w}}^{(0)} \hat{\mathbf{w}}^{(1)} \ldots \hat{\mathbf{w}}^{(L)} \right]$ and $\hat{\mathbf{w}}^{(l)}$ denotes the corresponding weights of the reconstructed network. Note that we do not normalize the error by the weight's magnitude. Next, the feature map distillation loss, $\mathcal{L}_{FMD}$, introduced in Romero et al. (2015), is defined by

$$\mathcal{L}_{FMD} = \frac{1}{|\mathcal{B}|} \sum_{i \in \mathcal{B}} \sum_l \|\mathbf{a}_i^{(l)} - \hat{\mathbf{a}}_i^{(l)}\|_2, \tag{4}$$

where $\mathbf{a}_i^{(l)}$ and $\hat{\mathbf{a}}_i^{(l)}$ are the $\ell_2$ normalized feature maps generated from the $i$-th sample in the mini-batch $\mathcal{B}$ at the $l$-th layer for the original and reconstructed networks, respectively. Finally, the knowledge distillation loss, $\mathcal{L}_{KD}$, from Hinton et al. (2014) is defined as

$$\mathcal{L}_{KD} = \frac{1}{|\mathcal{B}|} \sum_{i \in \mathcal{B}} \mathrm{KL} \left( \mathbf{a}_i^{(out)}, \hat{\mathbf{a}}_i^{(out)} \right), \tag{5}$$

where $KL\left(\cdot, \cdot\right)$ is the Kullback–Leibler divergence, $\mathbf{a}_i^{(out)}$ and $\hat{\mathbf{a}}_i^{(out)}$ are output logits generated from the $i$-th sample in the minibatch $\mathcal{B}$ by the original and reconstructed networks, respectively.

**Stochastic sampling** Similarly to minibatch sampling used in standard stochastic optimization algorithms, in each training step of NeRN we predict all the reconstructed weights but optimize only on a minibatch of them. This allows us to support large neural networks and empirically shows better convergence even for small ones. We explored three stochastic sampling techniques - (1) entire random layer, (2) uniform sampling, where we uniformly sample coordinates from across the model, (3) magnitude-oriented, where we use uniform sampling with probability $p_{uni}$, and weighted sampling with probability $1 - p_{uni}$, where the probability is proportional to the individual weight's magnitude. In practice, we chose the third technique with $p_{uni} = 0.8$. Ablation results are presented in Section 4.6.

### 3.3 PROMOTING SMOOTHNESS

While videos, images, and 3D objects all have some inherent smoothness, this is not the case with the weights of a neural network. For example, while adjacent frames in a video are likely to be similar, there is clearly no reason for adjacent kernels of a pre-trained network to have similar values. We hypothesize that by introducing some form of smoothness between our kernels we can simplify the task for NeRN. We now present and discuss two different methods to incorporate such smoothness.

**Regularization-based smoothness** A naive approach to promoting smoothness in the weights of a neural network is to explicitly add a loss term in the training process of the original network that encourages smoothness. Interestingly, we show that one can successfully learn a smooth network with slightly inferior performance on the original task simply by adding the smoothness term,

$$\mathcal{L}_{smooth} = \sum_{l=0}^{L-1} \sum_{f=0}^{F_l-2} \sum_{c=0}^{C_l-2} \Delta_c \left( \mathbf{w}^{(l)}\left[f, c\right], \mathbf{w}^{(l)}\left[f+1, c\right] \right) + \Delta_c \left( \mathbf{w}^{(l)}\left[f, c\right], \mathbf{w}^{(l)}\left[f, c+1\right] \right), \tag{6}$$

where $\Delta_c$ stands for the cosine distance, $L$ stands for the number of layers in the network, and $F_l, C_l$ stand for the number of filters and channels in a specific layer respectively. In $1 \times 1$ layers, we use a $l_2$ distance instead of the cosine distance, since the weight kernels are scalars. While conceptually interesting, this approach requires modifying the training scheme of the original network and access to its training data, and may result in a degradation in accuracy due to the additional loss term.

**Permutation-based smoothness** To overcome the downside of regularization, we introduce a novel approach for achieving kernel smoothness, by applying permutations over the pre-trained model's weights. That is, we search for a permutation of the kernels that minimizes equation 6 without changing the actual weights. Recall that NeRN maps each coordinate to the corresponding kernel, so in practice we keep the order of weights in the original network, and only change the order in which NeRN predicts the kernels.

To solve the permutation problem we formalize it using graph theory. We denote the complete graph $G_l = (V_l, E_l)$, $\forall l \in [1, L]$, where each vertex in $G_l$ is a kernel in $\mathbf{w}^{(l)}$, and the edge between vertex $i$ and $j$ is the cosine distance between the $i$-th and $j$-th kernels in $\mathbf{w}^{(l)}$ (for $1 \times 1$ kernels we replace the cosine distance with $\ell_2$ distance). Now, the optimal reordering of the weights in layer $l$ is equivalent to the minimal-weight Hamiltonian path (a path that goes through all vertices exactly once) in $G_l$, which is precisely the traveling salesman problem (TSP) (Reinelt, 1994). While TSP is known to be NP-Hard, we propose to use an approximation to the optimal solution using a greedy solution. That is, for $G_l$, start from an arbitrary vertex in $V_l$ and recursively find the closest vertex $V_l$ that has not yet been visited, until visiting all the vertices. We evaluated this method in our experiments presented in the following section. We consider two variants of this approach:

*Cross-filter permutations.* In each layer, consider $\mathbf{w}^{(l)}$ as a list of all kernels in layer $l$. We calculate the permutation across the entire list. The disadvantage of this approach is the overhead of saving the calculated ordering to disk. For a layer with $F_l$ filters and $C_l$ kernels each, the overhead is $F_l \cdot C_l \cdot \log_2 (F_l \cdot C_l) = F_l \cdot C_l \cdot (\log_2 F_l + \log_2 C_l)$ bits. For standard ResNet variants, saving these permutations entails a 4%-6% size overhead.

*In-filter permutations.* To reduce the overhead of the cross-filter permutations, we introduce in-filter permutations. For layer $l$, we first calculate the permutation of kernels inside each filter independently and then compute the permutation of the permuted filters. Figure 2 demonstrates this process. The overhead for a layer with $F_l$ filters and $C_l$ kernels each is $F_l \cdot C_l \cdot \log_2 C_l + F_l \cdot \log_2 F_l$ bits. For standard ResNet variants, saving these permutations entails a 2%-3% size overhead. Additional details are presented in appendix D. Intuitively, we'd expect the cross-filter permutations to be superior. Since we adopt a greedy algorithm this is not guaranteed, as will be shown later.

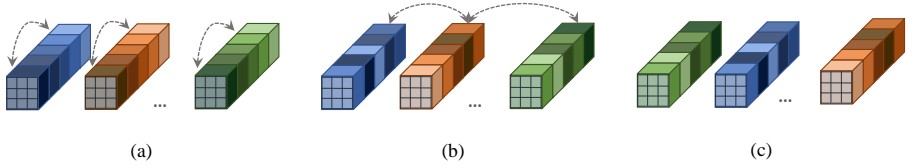

(a)    (b)    (c)

Figure 2: The process of applying **in-filter** permutations for a specific layer. After applying the permutations, each weight kernel is still located in its original filter. **(a)** The original filters in the layer. **(b)** The weight kernels within each filter are permuted. **(c)** The entire filters are permuted.

## 4 EXPERIMENTS

In this section we evaluate our proposed method on three standard vision classification benchmarks - CIFAR-10, CIFAR-100 (Krizhevsky et al., 2009) and ImageNet (Deng et al., 2009). For all benchmarks, we use NeRN to predict the weights of the ResNet (He et al., 2015a) architectures. The ResNet architectures were chosen for (1) their popularity, (2) their non-trivial design (deep layers, which incorporate $1 \times 1$ convolutions), and (3) their relatively high accuracy.

For every benchmark, we examine various NeRN hidden layer sizes and show the effectiveness of promoting smoothness in the original network's weights via weight permutations. For CIFAR-10, we present complete results and additionally explore the smoothness regularization method. For CIFAR-100 and ImageNet, we show only the best setup due to space constraints, and provide the complete results in Appendix E. Each experiment is executed with 4 different random seeds, where the mean and confidence intervals of the results are listed. We adopt the Ranger (Wright, 2019) optimizer, using a learning rate of $5 \cdot 10^{-3}$ and a cosine learning rate decay. The input coordinates are mapped to positional embeddings of size 240. The rest of the training scheme is presented in each individual section. We run our experiments using PyTorch on a single Nvidia RTX3090. The CIFAR experiments take about 1-3 hours, depending on the sizes of the original model and NeRN. ImageNet experiments take about 20-30 hours, depending on NeRN's size.

Following these, we discuss the idea of training NeRN without promoting any kind of smoothness to the original network, examine NeRN's ability to learn using noise images instead of the task data, and finally present some relevant ablation experiments.

Table 1: NeRN CIFAR-10 reconstruction results

| Architecture (Accuracy %) | Learnable Weights Size [MB] | Permutation Smoothness | NeRN Predictor | | Reconstructed Accuracy % |
|---|---|---|---|---|---|
| | | | Hidden Size | Model Size [MB] | |
| ResNet20 (91.69) | 1.03 | In-filter | 140 | 0.36 | 89.65 ± 0.33 |
| | | | 160 | 0.45 | 90.76 ± 0.13 |
| | | | 180 | 0.54 | 91.24 ± 0.08 |
| | | | 200 | 0.65 | 91.57 ± 0.09 |
| ResNet20 (91.69) | 1.03 | Cross-filter | 140 | 0.36 | **90.39 ± 0.05** |
| | | | 160 | 0.45 | **91.04 ± 0.07** |
| | | | 180 | 0.54 | **91.43 ± 0.08** |
| | | | 200 | 0.65 | **91.68 ± 0.06** |
| ResNet56 (93.52) | 3.25 | In-filter | 240 | 0.89 | 91.32 ± 0.07 |
| | | | 280 | 1.17 | 92.26 ± 0.12 |
| | | | 320 | 1.48 | 92.68 ± 0.05 |
| | | | 360 | 1.83 | 93.11 ± 0.08 |
| ResNet56 (93.52) | 3.25 | Cross-filter | 240 | 0.89 | **91.79 ± 0.14** |
| | | | 280 | 1.17 | **92.45 ± 0.11** |
| | | | 320 | 1.48 | **92.86 ± 0.09** |
| | | | 360 | 1.83 | **93.15 ± 0.12** |

Figure 3: NeRN ResNet56 reconstruction results on CIFAR-10 using different smoothness regularization factors, shown for the original model and the reconstructed one. The figures display the performance of NeRN with different hidden sizes, 240 on the left and 280 on the right.

## 4.1 CIFAR-10

For these experiments, we start by training ResNet20/56 to be used as the original networks. These ResNet variants are specifically designed to fit the low input size of CIFAR by limiting the input downsampling factor to ×4. We train NeRN for ~70k iterations, using a task input batch size of 256. In addition, a batch of $2^{12}$ reconstructed weights for the gradient computation is sampled in each iteration. Results are presented in Table 1. As expected, increasing the predictor size results in significant performance gains.

**Regularization-Based Smoothness** Here we show the results of applying smoothness via a regularization term on the original network training. Since incorporating an additional loss term results in a degradation in accuracy, there is an inherent tradeoff between the original network's accuracy and NeRN's ability to reconstruct the network. An optimal regularization factor balances the two,

Table 2: NeRN CIFAR-100 reconstruction results

| Architecture (Accuracy %) | Learnable Weights Size [MB] | Permutation Smoothness | NeRN Predictor | | Reconstructed Accuracy % |
|---|---|---|---|---|---|
| | | | Hidden Size | Model Size [MB] | |
| ResNet56 (71.35) | 3.25 | Cross-filter | 320 | 1.48 | $69.30 \pm 0.36$ |
| | | | 360 | 1.83 | $70.31 \pm 0.20$ |
| | 3.25 | In-filter | 400 | 2.22 | $70.97 \pm 0.14$ |

Table 3: NeRN ImageNet reconstruction results

| Architecture (Top-1/Top-5%) | Learnable Weights Size [MB] | Permutation Smoothness | NeRN Predictor | | Reconstructed Top-1 % | Reconstructed Top-5 % |
|---|---|---|---|---|---|---|
| | | | Hidden Size | Model Size [MB] | | |
| ResNet18 (69.76/89.08) | 41.91 | Cross-filter | 1024 | 12.99 | $67.55 \pm 0.05$ | $87.82 \pm 0.07$ |
| | | | 1140 | 15.97 | $68.21 \pm 0.12$ | $88.27 \pm 0.05$ |
| | | | 1256 | 19.27 | $68.74 \pm 0.03$ | $88.57 \pm 0.03$ |
| | | | 1372 | 22.87 | $69.07 \pm 0.05$ | $88.79 \pm 0.05$ |

achieving a high absolute reconstructed accuracy. Figure 3 demonstrates this using several regularization factors and two NeRN configurations. Using a hidden size of 240, the optimal is $5 \cdot 10^{-6}$, while using a hidden size of 280, the optimal is $1 \cdot 10^{-6}$. Complete results appear in Appendix E.1.

## 4.2 CIFAR-100

Here we start by training ResNet56 to be used as the original model. The setup is similar to that of CIFAR-10, only we train NeRN for $\sim$90k iterations. These experiments show similar trends, where promoting smoothness and using larger predictors results in better reconstruction. Results are presented in Table 2. Note that since this task is more complex than CIFAR-10, it requires a slightly larger NeRN for the same reconstructed architecture. Complete results appear in Appendix E.2.

## 4.3 IMAGENET

Here we show the flexibility of NeRN by learning to represent the ImageNet-pretrained ResNet18 from torchvision Paszke et al. (2019). Thanks to our permutation-based smoothness, which is applied post-training, NeRN can learn to represent a network that was trained on a large-scale dataset even without access to the training scheme of the original model. For ResNet-18, we predict only the $3 \times 3$ convolutions in the network which constitute more than $98\%$ of the entire convolutional parameters, while skipping the first $7 \times 7$ convolution and the downsampling layers. We train NeRN for 160k iterations, using a task input batch size of 32 (4 epochs). In addition, a batch of $2^{16}$ reconstructed weights for the gradient computation is sampled in each iteration. Results are presented in Table 3, where for evaluation we use the script provided by Wightman (2019). Here, interestingly, we require a relatively smaller NeRN. For example, the bottom row shows satisfying results, using a NeRN of $\sim 54\%$ the size of the original model. Complete results appear in Appendix E.3.

## 4.4 DATA-FREE TRAINING

In the previous experiments, NeRN was trained with images from the training data of the original network. Ideally, we would like to be able to reconstruct a network without using the original task data. This allows for a complete detachment of the original task when training NeRN. However, distilling knowledge using out-of-domain data is a non-trivial task. This is evident in the recent experiments by Beyer et al. (2022), where distilling knowledge using out-of-domain data achieved significantly worse results than in-domain data. Consequently, one would assume distilling knowledge from noise inputs should prove to be an even more difficult task, as the extracted features might not carry a meaningful signal. Interestingly, we show that given our combined losses and

method, NeRN achieves good reconstruction results without any meaningful input data, i.e by using uniformly sampled noise $X \sim U[-1, 1]$. Results are presented in Table 4.

## 4.5 RECONSTRUCTING NON-SMOOTH NETWORKS

Although our presented method for permutation smoothness offers a very small overhead, one might consider the possibility of reconstructing a model without promoting any kind of smoothness. Results are presented in table 5. Note that although NeRN is able to reconstruct non-smooth networks, the results are inferior to those of reconstructing smooth ones. While promoting smoothness improves results across all experiments, the accuracy gap is more significant for smaller predictors. In these cases, the predictor typically has lower capacity to capture non-smooth signals.

Table 4: NeRN CIFAR-10 reconstruction results using noise as input compared to omitting the distillation losses. Experiments were run using cross-filter permutations.

| Architecture (Accuracy %) | NeRN Hidden Size | Inputs | ↑ Accuracy % |
|---|---|---|---|
| ResNet20 (91.69) | 160 | ✗ | $88.36 \pm 0.39$ |
| | | Noise | $89.08 \pm 0.34$ |
| | 180 | ✗ | $90.21 \pm 0.32$ |
| | | Noise | $90.64 \pm 0.22$ |
| ResNet56 (93.52) | 280 | ✗ | $90.93 \pm 0.28$ |
| | | Noise | $91.40 \pm 0.37$ |
| | 320 | ✗ | $92.06 \pm 0.11$ |
| | | Noise | $92.42 \pm 0.06$ |

Table 5: NeRN CIFAR-10 reconstruction results for non-smooth networks.

| Architecture (Accuracy %) | NeRN Hidden Size | ↑ Accuracy % |
|---|---|---|
| ResNet20 (91.69) | 140 | $87.95 \pm 0.37$ |
| | 160 | $89.90 \pm 0.26$ |
| | 180 | $90.82 \pm 0.07$ |
| ResNet56 (93.52) | 240 | $87.54 \pm 0.24$ |
| | 280 | $90.83 \pm 0.15$ |
| | 320 | $92.11 \pm 0.07$ |

## 4.6 ABLATION EXPERIMENTS

We present a few ablation experiments for NeRN's loss functions in Table 6 & Figure 4. The results emphasize the importance of the distillation losses, allowing for a better and more stable convergence, and the need for a reconstruction loss. In addition, Table 7 examines the weight sampling methods for gradient computation as discussed in Section 3.2.

Table 6: Loss ablation results for NeRN ResNet20 reconstruction on CIFAR-10. We use a setup of cross-filter permutations and a hidden size of 160.

| $\mathcal{L}_{recon}$ | $\mathcal{L}_{KD}$ | $\mathcal{L}_{FMD}$ | ↑ Accuracy % |
|---|---|---|---|
| ✓ | ✗ | ✗ | $88.36 \pm 0.39$ |
| ✗ | ✓ | ✓ | $10.30 \pm 0.48$ |
| ✓ | ✓ | ✓ | $\mathbf{91.04 \pm 0.07}$ |

Table 7: Weight sampling ablation results. Setup similar to Table 6

| Weight Sampling | ↑ Accuracy % |
|---|---|
| All Weights | $90.99 \pm 0.13$ |
| Random Layer | $73.77 \pm 5.49$ |
| Random Batch | $91.00 \pm 0.08$ |
| Random Batch (weighted) | $\mathbf{91.04 \pm 0.07}$ |

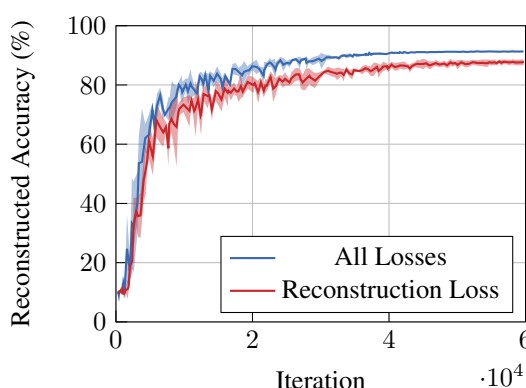

Figure 4: NeRN ResNet20 training plot on CIFAR-10. Setup similar to Table 6. Incorporating the distillation losses allows for a faster and more stable convergence.

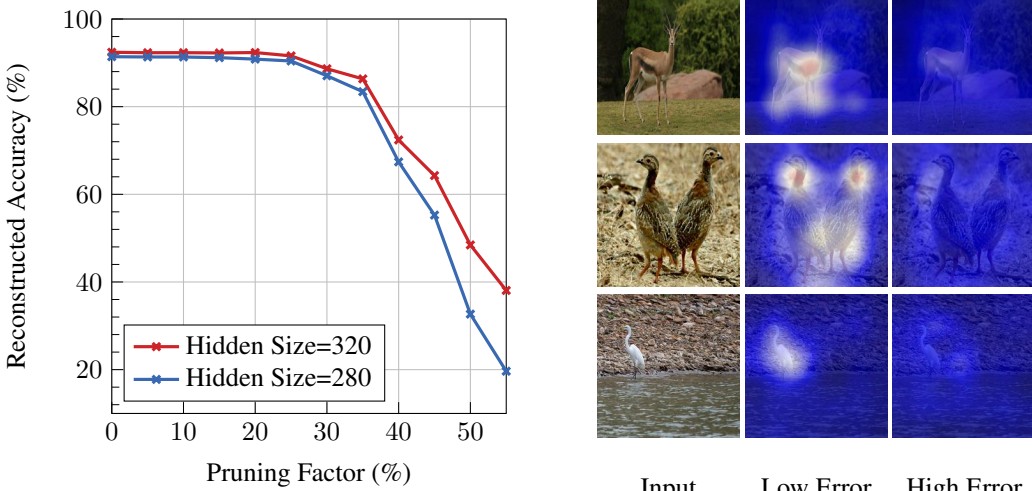

Figure 5: Two additional applications for NeRN. The left figure presents naive unstructured magnitude pruning results. Pruning was applied on a NeRN trained to reconstruct ResNet56 on CIFAR-10, using cross-filter permutations. The right figure presents the averaged activations of 10 *original* filters with the lowest/highest reconstruction error in *layer4.0.conv2* of ResNet18 trained on ImageNet. For this, we used a NeRN trained with in-filter permutations and a hidden size of 1256.

## 5   ADDITIONAL APPLICATIONS

NeRNs offer a new viewpoint on neural networks, encoding the network weights themselves in another network. We believe that various research directions and applications can arise from this new representation, and below we examine some possible applications that can benefit from NeRNs.

**Weight Importance**  Through the distillation losses guiding the optimization process, we hypothesize that NeRN prioritizes the reconstruction of weights based on their influence on the activations and logits. This observation means that NeRN implicitly learns weight importance in a network-global manner. By extracting this information from NeRN we can visualize important filters, which will be those with the lowest relative reconstruction error. Figure 5 visualizes the average activation map of the 10 filters with the lowest/highest reconstruction error in a specific ResNet18 layer. The filters with the lower reconstruction error do indeed correspond to high valued activation maps.

**Meta-Compression**  NeRN offers a compact representation of the original network, which is a neural network by itself. We propose that one can compress the NeRN predictor to achieve a more disk-size economical representation. To demonstrate this, we apply naive magnitude-based pruning on the predictor. The results are presented in Figure 5. As further extensions one can also examine more sophisticated compression techniques e.g., structured pruning and quantization.

Another interesting usage is the NeRN of an already pruned network to further reduce its disk size. That is a promising direction for two reasons, (1) NeRN predicts only weights, and having less weights to predict simplifies the task, and likely also the size of NeRN, and (2) NeRN can be used post-training, without access to the task data.

## 6   CONCLUSION

We propose a technique to learn a neural representation for neural networks (NeRN), where a predictor MLP, reconstructs the weights of a *pretrained* CNN. Using multiple losses and a unique learning scheme, we present satisfying reconstruction results on popular architectures and benchmarks. We further demonstrate the importance of weight smoothness in the original network, as well as ways to promote it. We finish by presenting two possible applications for NeRN, (1) weight importance analysis, where the importance is measured by NeRN's accuracy, and (2) meta-compression, where the predictor is pruned to achieve a disk-size compact representation, possibly without data.

**Reproducibility Statement**   An important aspect for the authors of this paper is code usability. We have developed a generic and scalable framework for NeRN, which we provide in the supplementary materials. The significant information for reproducing the experiments in this paper is listed in Section 3. In addition, we provide the relevant configuration files and a README file containing instructions in the supplementary material. We hope this allows the reader to reproduce the results in an accessible manner.

**Acknowledgements**   We would like to thank Yoav Miron, Gal Metzer and Yuval Alaluf for their valuable insights throughout the research and writing of this paper. This research was supported by The Israel Science Foundation (grant No. 1589/19), and in part by the Israeli Council for Higher Education (CHE) via the Data Science Research Center, BGU, Israel.

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

# Appendices

## A   VISUALIZING RECONSTRUCTED KERNELS

To qualitatively demonstrate NeRN's reconstruction, we train NeRN to reconstruct an ImageNet-pretrained ResNet18, and visualize the reconstructed kernels. For this experiment, we use a hidden size of 1256. Figure 6 presents the original and reconstructed kernels of a specific channel in the second layer of ResNet18, using a $8 \times 8$ grid of $3 \times 3$ kernels.

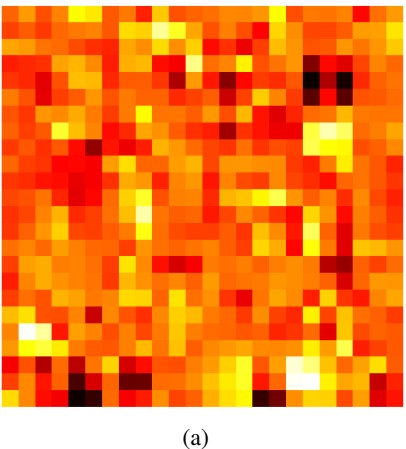
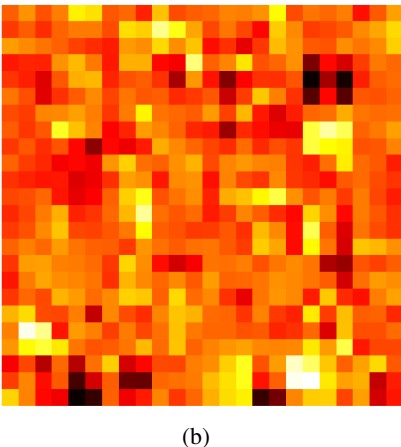

(a)                                                           (b)

Figure 6: An $8 \times 8$ grid visualization of part of the original and reconstructed $3 \times 3$ kernels in the second layer of ResNet18, trained on ImageNet. (a) are the original kernels, (b) are the reconstructed kernels.

## B   NERN INITIALIZATION

A standard neural network initialization method (He et al., 2015b) aspires to preserve the activation's variance throughout the network. Due to its non-standard outputs, we initialize NeRN such that the initially predicted weights are of similar mean and variance to the original model's weights. This is done using a similar initialization method to that of HyperNetworks (Ha et al., 2016). We empirically found this initialization allows for relatively faster convergence.

## C   CONVOLUTIONAL PARAMETERS SIZE

Table 8: Size of convolutional parameters in standard ResNet architectures.

| Architecture | Task | Total Size [MB] | Convolutional Parameters Size [MB] | Convolutional Parameters Size (%) |
|---|---|---|---|---|
| ResNet20 | CIFAR-10 | 1.04 | 1.03 | 99.04% |
| ResNet56 | CIFAR-10 | 3.26 | 3.25 | 99.69% |
| ResNet20 | CIFAR-100 | 1.06 | 1.03 | 97.17% |
| ResNet56 | CIFAR-100 | 3.29 | 3.25 | 98.78% |
| ResNet18 | ImageNet | 44.59 | 42.60 | 95.54% |

# D PERMUTATION-BASED SMOOTHNESS

## D.1 SIZE OVERHEAD

As discussed in Section 3.3, saving the original model's weight permutations for a layer with $F_l$ filters and $C_l$ kernels each, weighs either $F_l \cdot C_l \cdot (\log_2 F_l + \log_2 C_l)$ bits for the cross-filter permutations variant, or $F_l \cdot C_l \cdot \log_2 C_l + F_l \cdot \log_2 F_l$ bits for the in-filter permutations variant. Table 9 presents this size overhead for both variants on the architectures we experimented on throughout the paper.

Table 9: Size overhead for weight permutations on standard ResNet architectures.

| Architecture | Task | Total Size [MB] | In-channel Permutations | | Cross-channel Permutations | |
|---|---|---|---|---|---|---|
| | | | Size [MB] | Overhead (%) | Size [MB] | Overhead (%) |
| ResNet20 | CIFAR-10 | 1.04 | 0.02 | 1.92 | 0.04 | 3.85 |
| ResNet56 | CIFAR-10 | 3.26 | 0.065 | 1.99 | 0.128 | 3.93 |
| ResNet20 | CIFAR-100 | 1.06 | 0.02 | 1.89 | 0.04 | 3.77 |
| ResNet56 | CIFAR-100 | 3.29 | 0.065 | 1.98 | 0.128 | 3.89 |
| ResNet18 | ImageNet | 44.59 | 1.246 | 2.79 | 2.505 | 5.62 |

## D.2 PERMUTATIONS ILLUSTRATION

In the permutation figures, the color of the weight kernels signify the filters they belong to originally, **not** the value of the weights. The in-filter permutation process is demonstrated in figure 2. The cross-filter permutation process is demonstrated in figure 7

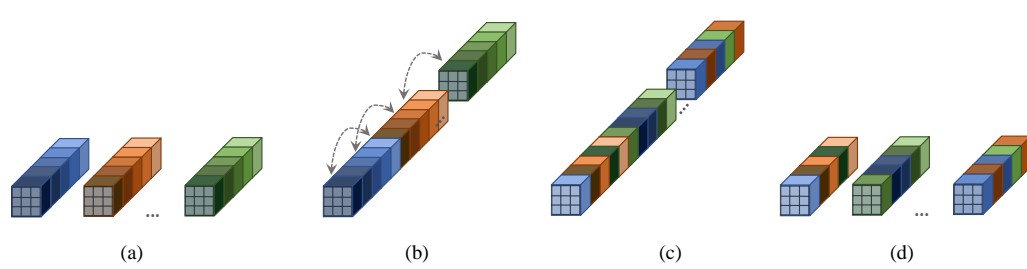

Figure 7: The process of applying **cross-filter** permutations for a specific layer. After applying the permutations, the weight kernels might not be located in their original filters. **(a)** The original filters in the layer. **(b)** The filters are concatenated to a single tensor of weight kernels. **(c)** The weight kernels in the entire tensor are permuted. **(d)** The permuted tensor is reshaped into separate filters.

# E  COMPLETE RECONSTRUCTION RESULTS

## E.1  CIFAR10 WITH REGULARIZATION-BASED SMOOTHNESS

Table 10: NeRN CIFAR-10 reconstruction results using regularization-based smoothness on ResNet56. Learnable weights size is 3.25 MB. Notice the tradeoff between original network accuracy, and NeRN's ability to reconstruct the network.

| Smoothness Regularization Factor | Original Accuracy % | NeRN Predictor Hidden Size | Model Size [MB] | Reconstructed Accuracy % |
|---|---|---|---|---|
| $1 \cdot 10^{-6}$ | 93.35 | | | $90.14 \pm 0.15$ |
| $5 \cdot 10^{-6}$ | 92.07 | 240 | 0.89 | $\mathbf{90.96 \pm 0.11}$ |
| $1 \cdot 10^{-5}$ | 91.55 | | | $90.82 \pm 0.03$ |
| $5 \cdot 10^{-5}$ | 90.35 | | | $89.82 \pm 0.05$ |
| $1 \cdot 10^{-6}$ | 93.35 | | | $\mathbf{91.74 \pm 0.10}$ |
| $5 \cdot 10^{-6}$ | 92.07 | 280 | 1.17 | $91.45 \pm 0.03$ |
| $1 \cdot 10^{-5}$ | 91.55 | | | $91.08 \pm 0.04$ |
| $5 \cdot 10^{-5}$ | 90.35 | | | $90.10 \pm 0.05$ |

## E.2  CIFAR100

Table 11: Complete NeRN CIFAR-100 reconstruction results

| Architecture (Accuracy %) | Learnable Weights Size [MB] | Permutation Smoothness | NeRN Predictor Hidden Size | Model Size [MB] | Reconstructed Accuracy % |
|---|---|---|---|---|---|
| ResNet56 (71.35) | 3.25 | None | 320 | 1.48 | $68.76 \pm 0.08$ |
| | | | 360 | 1.83 | $70.09 \pm 0.06$ |
| | | | 400 | 2.22 | $70.83 \pm 0.13$ |
| ResNet56 (71.35) | 3.25 | In-filter | 320 | 1.48 | $68.92 \pm 0.12$ |
| | | | 360 | 1.83 | $70.30 \pm 0.20$ |
| | | | 400 | 2.22 | $\mathbf{70.97 \pm 0.14}$ |
| ResNet56 (71.35) | 3.25 | Cross-filter | 320 | 1.48 | $\mathbf{69.30 \pm 0.36}$ |
| | | | 360 | 1.83 | $\mathbf{70.31 \pm 0.20}$ |
| | | | 400 | 2.22 | $70.86 \pm 0.17$ |

## E.3  IMAGENET

Table 12: NeRN ImageNet reconstruction results using ResNet18

| Architecture (Top-1/Top-5%) | Learnable Weights Size [MB] | Permutation Smootheness | NeRN Predictor Hidden Size | Model Size [MB] | Reconstructed Top-1 % | Reconstructed Top-5 % |
|---|---|---|---|---|---|---|
| ResNet18 (69.76/89.08) | 41.91 | In-filter | 1024 | 12.99 | $67.48 \pm 0.06$ | $87.78 \pm 0.05$ |
| | | | 1140 | 15.97 | $\mathbf{68.25 \pm 0.03}$ | $88.29 \pm 0.04$ |
| | | | 1256 | 19.27 | $68.71 \pm 0.09$ | $88.54 \pm 0.02$ |
| | | | 1372 | 22.87 | $69.03 \pm 0.02$ | $88.72 \pm 0.02$ |
| ResNet18 (69.76/89.08) | 41.91 | Cross-filter | 1024 | 12.99 | $\mathbf{67.55 \pm 0.05}$ | $87.82 \pm 0.07$ |
| | | | 1140 | 15.97 | $68.21 \pm 0.12$ | $88.27 \pm 0.05$ |
| | | | 1256 | 19.27 | $\mathbf{68.74 \pm 0.03}$ | $88.57 \pm 0.03$ |
| | | | 1372 | 22.87 | $\mathbf{69.07 \pm 0.05}$ | $88.79 \pm 0.05$ |

Table 13: NeRN ImageNet reconstruction results using SqueezeNet

| Architecture (Top-1 %) | Learnable Weights Size [MB] | Permutation Smootheness | NeRN Predictor | | Reconstructed Top-1 % |
|---|---|---|---|---|---|
| | | | Hidden Size | Model Size [MB] | |
| SqueezeNet (58.19) | 2.74 | In-filter | 340 | 1.48 | $56.94 \pm 0.05$ |
| | | | 340 | 1.65 | $57.44 \pm 0.05$ |
| | | | 360 | 1.83 | $57.64 \pm 0.07$ |

## F  A NOTE ON POSITIONAL EMBEDDINGS

As NeRN learns to represent kernels by mapping from the positional embedding of a given coordinate to its corresponding kernel, it is clear that the embeddings we choose also play a significant part in the learning process. As adjacent convolutional kernels are relatively similar to one another after we promote smoothness, one might consider the possibility of creating similarly behaving positional embeddings. These positional embeddings should be, on the one hand, slowly changing with respect to adjacent kernel coordinates, and on the other hand, highly separable with respect to distant coordinates. In practice, this can be achieved by finding the right basis for our positional embedding, as shown in Figure 8. Although we had hypothesized that incorporating this inductive bias might assist NeRN in reconstructing smooth networks, numerous experiments have refuted this theory. Nevertheless, further investigation into the chosen positional embeddings is needed, and we leave this as a topic for future research.

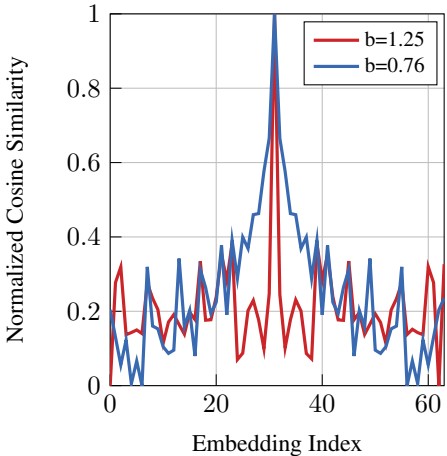

Figure 8: Normalized cosine similarity between positional embeddings of $(0, 0, c), c \in [0, 63]$ to embedding of coordinate $(0, 0, 31)$, using different base frequencies.

