# OpenReview forum: "NeRN: Learning Neural Representations for Neural Networks"
_ICLR.cc/2023/Conference — ICLR 2023 notable top 25%_

### Official Review · Reviewer_HBhL · 2022-10-24

**Confidence:** 5
**Correctness:** 3
**Technical Novelty And Significance:** 4
**Empirical Novelty And Significance:** 4
**Recommendation:** 8

**Clarity, Quality, Novelty And Reproducibility:**

The paper provides a novel method for NeRN, which surprisingly reconstruct the weights of CNNs with implicit neural representations. The paper also shows some promising applications of NeRN. Overall, in my opinion, the paper is novel and the idea of weight reconstruction is interesting. It would be a good paper that provides a possible solution to hyper-networks or meta-learning.


**Strength And Weaknesses:**

Strengths:
(1)	The paper provides a novel method for NeRN, which surprisingly reconstruct the weights of CNNs with implicit neural representations.
(2)	To boost the performance, the paper also introduces a smoothness constraint on NeRN.
(3)	The paper shows two promising applications of NeRN, including weight importance visualization and meta-compression.


Weaknesses:
(1)	In the meta-compression, it is unclear how the paper performs meta-compression. The paper claims “we apply naïve magnitude-based pruning on the predictor”, why do not directly prune raw pre-trained weights? I do not understand how the reconstructed weight works. Given a NeRN, we can map coordinates to weights and can also filter large magnitudes or map to sparse structured weights. If so, how to select the channel number for different layers? The paper claims the NeRN may reduce the disk size. A drawback could increase computational cost since we must reconstruct weights each forward. I am not sure about the inference time compared with the original weights. Moreover, because it needs to reconstruct intermediate temporary networks, how does it reduce the disk size? It would be better if some more applications are introduced.

(2)The paper mainly focuses on “weight generator”, while the reference [r1] focuses on “weight discriminator”. Is it possible to design a “weight GAN” to generate realistic weights (sharp and containing high frequencies) without smoothness constraints? Or some permutation constraints (cross-filter and in-filter, e.g., chain normalization rule [r1]) on weights can be exploited as extra constraints to boost the reconstruction instead of strong smoothness assumptions? Some discussions would be nice if possible.

[r1]Understanding Weight Similarity of Neural Networks via Chain Normalization Rule and Hypothesis-Training-Testing


**Summary Of The Paper:**

This paper proposes a neural representation method of neural networks (NeRN) to represent the weights of neural networks, which maps coordinates to convolutional kernels. The paper also considers spatial smoothness constraints on networks’ weights  to help NeRN. Moreover, the paper also points out two applications using NeRN, including filter visualization and network structure pruning. Experiments show good reconstructions of CNNs.

**Summary Of The Review:**

(1) A novel method for NeRN for weight reconstruction of CNNs.
(2) Good experimental results of reconstructed weights of neural networks.
(3) Some weaknesses exist, but not a big question.

---

> ### Author Response · Authors · 2022-11-14
> **Response to Reviewer HBhL**
>
> Dear reviewer HBhL, we thank you for your time and constructive review.
>
> 1. **Regarding meta-compression:** We suggest meta-compression as an application for NeRN, where one compresses NeRN to achieve an economic neural representation of the original network. In addition, we mention that one can train NeRN to predict an already compressed network, and then additionally compress NeRN. This means that there is no problem in pruning the pre-trained weights as you suggested, and compressing NeRN is a complementary method to pruning the original network. That is, the training of NeRN does not include the pruning or compression of the weights, but instead, NeRN will represent the already compressed network, and the number of channels/filters or sparsity pattern in different layers will be determined by the compression algorithm applied to the original network. Incorporating the compression of the network during the training of NeRN is an interesting idea that we might look into in the near future. We would also like to mention that we do not treat the compression of NeRN as a competitor to SOTA compression techniques, but rather as an additional complementary method, for which we provide a proof of concept by applying naive magnitude pruning on NeRN. Compressing NeRN will achieve a disk-size economical representation. It will not require saving any additional temporary weights, since the weights predicted by NeRN are only loaded to the RAM for running inference.
>
>     Regarding the latency in running inference, the weight prediction step occurs only once when loading the weights, and will be used later for any number of forward passes. This means that one does not have to reconstruct the weights for every inference. For ResNet20/ResNet56 it takes a few seconds to load the positional embeddings and predict the weights. For ResNet18, it takes around 40 seconds for the entire process.
>
> 2. **Regarding a “weight GAN”:** This is an interesting concept. Table 5 in our paper shows the results of our method, without applying any smoothness constraints. Even though the accuracy of the reconstructed network is worse than the one achieved by inducing smoothness, we still manage to achieve relatively good accuracy. In this case, the weights have some high frequencies (as the raw pre-trained network). In addition, there are a few weight generation papers we referred to in the related work, for example “Parameter prediction for unseen deep architectures. (NeurIPS 2021)” which might be able to generate these realistic weights without any smoothness constraints.
>
>     Regarding the referenced paper: Thank you for the reference. It provides an interesting technique for measuring weight similarity of networks by leveraging the fact that the weights can be permuted while achieving the same outcome. Intuitively, this method can be used as a replacement for our reconstruction loss, thus allowing NeRN to deviate from the exact pretrained weight tensor while still achieving similar accuracy. It might also allow for a relaxation in the smoothness requirements. It is important to note that the reconstructed network might result in significantly different weight tensors (l2-wise) than the original network. On the one hand, it contradicts the idea of our implicit representation, but on the other hand, it might lead to very interesting directions since different seeds might result in different reconstructed networks that might be ensembled in some way.

---

### Official Review · Reviewer_uYwd · 2022-10-24

**Confidence:** 4
**Correctness:** 4
**Technical Novelty And Significance:** 4
**Empirical Novelty And Significance:** 2
**Recommendation:** 6

**Clarity, Quality, Novelty And Reproducibility:**

The clarity, quality, and novelty and reproducibility are strong. I quickly inspected some of the code and it looks good. The only thing missing is comparison with other methods.

**Strength And Weaknesses:**

The technique seem novel and interesting. The paper in itself is sound and thorough. I especially like the smoothness regularization and experiments around it, e.g. the positional encodings. The weakness is the comparison in performance to other related work, like knowledge distillation etc.

**Summary Of The Paper:**

This paper present NERN which is a technique for learning neural representations for convolutional NNs. The parametrize the weight space by (layer, filter, channel) and train a MLP to predict the  k x k convolutional kernels. They combine different loss functions, experiments on CIFAR and ImageNet as well as provide multiple ablation studies.

**Summary Of The Review:**

Great paper. I only miss comparison with other methods. Would love to hear authors comments on this.

---

> ### Author Response · Authors · 2022-11-14
> **Response to Reviewer uYwd**
>
> Dear reviewer uYwd, we thank you for your time and constructive review.
>
> We understand your comment and agree that when working in a previously defined and explored domain, one must provide comparisons. To the best of our knowledge, the task of reconstructing a network’s pretrained weights using a different network was not explored in previous literature, especially when treating the task as a neural representation problem. We have provided some examples of previous weight prediction papers in the related work section of the paper, but these do not attempt to reconstruct a set of pretrained weights, so sadly, we can not trivially adapt these methods to solve this task. Thus, comparing them with our method is problematic. For example, some of these papers tackle the task of initializing weights for a given architecture, perhaps on unseen architectures. These methods use the classical approach, where a training set of architectures and their weights are defined, and the trained model generalizes to unseen architectures, creating starting points for fine-tuning the weights. On the other hand, we attempt to represent the weights of a specific pretrained network using an implicit neural representation.
>
> **Regarding knowledge distillation:** We incorporate knowledge distillation in our paper in order to guide NeRN's training scheme. NeRN’s loss term incorporates not only the weight reconstruction error but also the feature map reconstruction error and the KL-divergence between the reconstructed and original logits. Knowledge distillation is generally used as a method for training a smaller model by mimicking the performance of a larger model. In our work, we do not attempt to create a smaller model that solves the original task but rather a compact representation of the original network’s weights. In other words, the approach of NeRN is about the neural representation of a network’s weights, for various purposes, and knowledge distillation aims at distilling knowledge for the purpose of compression. One might use these methods jointly, by first applying knowledge distillation for compression, and later train NeRN to predict the compressed network (and even further compress NeRN itself, as we demonstrate in the applications section). Thus, it is not possible to directly compare our method to knowledge distillation in general. We did explore the effectiveness of incorporating knowledge distillation losses in NeRN — see the ablation studies in Table 6 and Figure 4 in our paper. The studies demonstrate that most of the heavy lifting in the training process is done by the reconstruction loss, and the distillation losses help in stabilizing the training and achieving slightly better results.
>
> We believe the task presented in our paper is novel, and even though we couldn’t find any work which attempts to solve it, we would of course appreciate any suggestions or referrals to papers that we could compare to.

---

### Official Review · Reviewer_MGd2 · 2022-10-27

**Confidence:** 3
**Clarity, Quality, Novelty And Reproducibility:** The paper is well-written and has som…
**Correctness:** 3
**Technical Novelty And Significance:** 2
**Empirical Novelty And Significance:** Not applicable
**Recommendation:** 6

**Strength And Weaknesses:**

Strengths:
1. The problem is interesting and well-motivated. The authors point out that by changing the order in which the neural network predicts the kernels, we can improve the smoothness of the weights and yield better reconstruction accuracy.
2. Extensive experiments. The authors show the effectiveness of several smoothness-enhancing techniques and validate them on various datasets.
Weakness:
1. It introduces both size and time overhead to apply permutation smoothness. I am wondering about the time complexity to find proper permutations. Is it scalable with the network size?
2. It would be better if the authors can provide their intuitions of why cross-filter permutation is in general the best strategy. Is this conclusion universal or dependent on the data the original network is trained on or dependent on the architecture of the original network?

**Summary Of The Paper:**

The authors have proposed several ways to enhance the smoothness of the reconstructed weights. They show that incorporating a smoothness constraint over the original network's weights aids the neural networks towards a better reconstruction. They illustrate the utility of smoothness in reconstructed weights by experimenting on several real datasets.

**Summary Of The Review:**

Overall, the paper is well-written and well-motivated. The experiments are extensive.

---

> ### Author Response · Authors · 2022-11-14
> **Response to Reviewer MGd2**
>
> Dear reviewer MGd2, we thank you for your time and constructive review.
>
> We would like to begin by mentioning that even though the permutation smoothness process improves NeRN’s reconstruction abilities, and is a part of our contributions, it is *not* our main contribution nor the core of our method. Our main idea is to encode a given network’s weights using a neural representation.
>
> 1. **Regarding the time overhead:** Computing the permutations is a one-time operation that occurs before training NeRN. As explained in the paper, the permutations are cached and later used to reconstruct the original network by changing the order in which the weights are predicted (technically, they are applied on the positional embeddings), which is negligible in terms of time.  On our CPU, computing the cross-filter permutations which achieved the best results for ResNet20/ResNet56 takes around 30 seconds. For ResNet18, computing the in-filter permutations which achieved the best results takes around 40 seconds. As expected, the cross-filter permutations are more complex and thus take a few hours to finish for ResNet18 (still a negligible time compared to training ResNets). Hence, this one-time operation does not constitute a significant overhead. Additionally, it is important to note that since the permutations are applied at each layer separately, this process can be easily parallelized across multiple CPUs, and the latency can be significantly improved. We are working on this feature in our code. We would also like to mention that the alternative method we presented, smoothness regularization training, requires no such overhead.
> 2. **Regarding the cross-filter permutations:** We use the permutations as a method for minimizing the smoothness term (Equation 6 in the paper). The cross-filter permutations add an additional degree of freedom to the in-filter permutations, allowing for weight kernel reordering between different filters, and thus intuitively might be superior. Empirically, we found that in most cases, cross-filter permutations perform better than in-filter permutations. However, as we explain in the paper, the cross-filter approach is not guaranteed to yield a lower smoothness loss since we adopt a greedy algorithm to find these permutations. In addition, even if a specific cross-filter permutation has a lower smoothness loss than another, it does not necessarily mean that applying it will bring NeRN to a better reconstruction (although it is highly correlated, as shown in our work). Since the weight kernel values are dependent on the original network architecture and task, you are correct to mention that it depends on these various factors. These observations are demonstrated in the ResNet18 reconstruction results, where using the in-filter permutations allowed for a better reconstruction.

---

### Official Review · Reviewer_jngE · 2022-10-27

**Confidence:** 4
**Correctness:** 4
**Technical Novelty And Significance:** 3
**Empirical Novelty And Significance:** 3
**Recommendation:** 8

**Clarity, Quality, Novelty And Reproducibility:**

- The paper is very clearly written and easy to understand except for some sections I have mentioned in the Weaknesses section.
- The quality of this paper is high. The experiments are thorough, and the discussion is compelling.
- I have not attempted to reproduce the results of this paper.

# Minor corrections
- line 4 on the second paragraph of page 5: wights should be weights.

**Strength And Weaknesses:**

# Strengths
- The paper is very well written and easy to read.
- The results are compelling.
- The method is novel to my knowledge.
- The approach considered suggests future explorations that could be useful.

# Weaknesses
- Some sections are not as clear as they could be. For instance, the section detailing the permutation-based smoothness was hard to understand and could be better explained, perhaps with the addition of an extra diagram.
- Some of the decisions are not extremely well motivated or ablated against. For instance:
  - What is the difference between using positional encodings and using directly using coordinates like those used in the SIREN?
  - Why is smoothness designed in this way? There doesn't seem to be any strong motivation for why neighboring kernels should look similar. What assumption is this smoothness based on?
- I would like some experiments with non-ResNet CNN architectures to be explored. Right now it's hard to say if the NeRN generalizes to any other architectures even though that is what is implicitly assumed by the paper.
- The performance of the NeRNs is still consistently lower than that of the original CNNs. What might be some reasons for this? It seems like increasing the capacity of the NeRN helps but the model seems to be limited in some way based on the results.
- How does this method compare to HyperNets which also attempt to predict the weights of a network albeit in a different way?

**Summary Of The Paper:**

In this paper, the authors present a novel method inspired by neural representations like NeRFs, SIRENs, and NeRVs to represent convolutional neural networks. They use position encodings as the input coordinates to for the 5-layer MLP NeRN to represent the location of the convolutional kernel within the architecture. the output of the NeRN is the parameters of the corresponding kernel. They also propose a hybrid objective function and some smoothing techniques to improve the model performance. Their approach is evaluated on 3 different ResNet architectures on CIFAR-10, CIFAR-100, and ImageNet and achieves comparable performance to the performance of the original architectures while using less space.

**Summary Of The Review:**

I think this is a really interesting paper that has the opportunity to lead to some further interesting research directions in the community.
However, there are some weaknesses I pointed out in the weaknesses section that I would appreciate if the authors could address.

---

> ### Author Response · Authors · 2022-11-14
> **Response to Reviewer jngE**
>
> Dear reviewer jngE, we thank you for your time and constructive review.
>
> 1. **Regarding the clarity of the permutation-based smoothness:** We agree with your observation on the usefulness of a diagram and were happy to create one. Please refer to the updated version of the paper, the figure appears in Appendix (D). We hope our proposed permutation process will now become clearer.
>
> 2. **Regarding the mentioned decisions:**
>     * Even though SIREN does not employ positional encodings for the input coordinates, it manages to produce great results by using a special periodic layer called the SineLayer. They feed the input through a linear layer and then apply a periodic sine function. We adopt a NeRF-like architecture and use the popular periodic positional encodings. This can be viewed as a SineLayer with a reverse order of operations at the first layer, i.e start by applying the periodic activation function. Using positional encodings became a very standard method for neural representations in recent years, thus we chose them. However, it might be interesting to try and use SIREN as the NeRN predictor. We do not believe this would change the expressiveness of the model, though might be worth further examination. We tried using the plain coordinates used in SIREN as input to the current architecture of NeRN, but all the experiments failed to converge.
>     * As explained in the paper, previous neural representation works focused on natural signals such as videos, images and 3D objects. All of these have a form of inherent smoothness. For example, similar frames in a video are likely to be similar and so do adjacent pixels in an image. We hypothesized that by introducing a form of smoothness to the weight kernels, the task could be simplified, which is in fact the case as shown by our ablation studies in the paper. Applying smoothness allowed for a better reconstructed accuracy and a lower variance in results between different runs. In addition, when the smoothed weights are complemented with “smoother” positional encodings, the best reconstruction results are obtained.
>
> 3. **Regarding different architectures:** We believe the different ResNet architectures are good representatives for the complexity of standard CNNs, thus we mainly focused on them in our experiments for the paper. In the camera-ready version of the paper, we will add a link to the code repository which will list additional results on other architectures. In the meantime, here are some preliminary results on SqueezeNet (1.1) for ImageNet using in-filter permutations:
>
>     |               | ImageNet Top-1 Accuracy (%) | Learnable Weights Size (MB) |
>     |:----------------------------------:|:---------------------------:|:--------------------------:|
>     | SqueezeNet 1.1  (Original Network) |          **58.19**          |          **2.74**          |
>     |       NeRN (Hidden Size 320)       |         56.94 ± 0.05        |            1.48            |
>     |       NeRN (Hidden Size 340)       |         57.44 ± 0.05        |            1.65            |
>     |       NeRN (Hidden Size 360)       |       **57.64 ± 0.07**      |          **1.83**          |
>
>     These results demonstrate NeRN’s effectiveness since SqueezeNet is a lightweight ImageNet architecture with less weight redundancy than large ResNet variants.
>
> 4. **Regarding the performance gap:** This is an interesting observation, which we did not cover in the paper due to space constraints. Even though the accuracies of the reconstructed networks are still slightly inferior to the baselines, increasing NeRN’s capacity allows for a better reconstruction, and when using a NeRN the size of the original network, we achieve an accurate representation, as demonstrated by the following results on ResNet56/CIFAR-10, using cross-filter permutations:
>
>     |           | CIFAR-10 Accuracy (%) | Learnable Weights Size (MB) |
>     |:---------------------------:|:---------------------:|:--------------------------:|
>     | ResNet56 (Original Network) |       **93.52**       |          **3.25**          |
>     |    NeRN (Hidden Size 280)   |      91.49 ± 0.14     |            1.17            |
>     |    NeRN (Hidden Size 320)   |      92.40 ± 0.07     |            1.48            |
>     |    NeRN (Hidden Size 360)   |      92.80 ± 0.14     |            1.83            |
>     |    NeRN (Hidden Size 400)   |      93.18 ± 0.09     |            2.22            |
>     |    NeRN (Hidden Size 440)   |      93.32 ± 0.01     |            2.64            |
>     |    NeRN (Hidden Size 492)   |    **93.53 ± 0.04**   |          **3.25**          |
>
>     Intuitively, NeRN is limited by the accuracy of the original network, but, at least according to this experiment, this upper bound is achievable by NeRN once we increase its capacity.

---

> > ### Author Response · Authors · 2022-11-14
> > **Response to Reviewer jngE (Cont.)**
> >
> > 5. **Regarding HyperNetworks:** Even though HyperNetworks tackle the task of predicting network’s weights, they are significantly different from NeRN in both implementation and problem domain. HyperNetworks do not attempt to predict the weights of a pretrained neural network, but are used as independent architectures that are trained “from scratch” using the original task loss and data. In contrast, NeRN is agnostic to the original task loss, and attempts to predict the set of previously defined weights. To the best of our knowledge, this is a novel task and we couldn’t find any related work which attempts or can trivially be adapted to solve the same task. This means we can not simply compare NeRN to HyperNetworks in terms of accuracy.
> >
> > 6. **Regarding the spelling error:** Thank you for the observation. It is now fixed.

---

### Decision · Program_Chairs · 2023-01-20

**Decision:**

Accept: notable-top-25%

**Justification For Why Not Higher Score:**

it would be unfair to the authors of the paper to state an arbitrary reason for not attributing a higher score. The paper has significant potential to be impactful. As a meta-reviewer who tries to be fair, objective and intellectually honest, the only reason for which the paper has not been considered for the spotlight or oral is that it has a grade not strictly greater than 7.

**Justification For Why Not Lower Score:**

Honestly, Recommending whether an accepted paper should be attributed to the mention of a poster, spotlight or an oral is very subjective if it does not follow fair and objective rules. A proxy to such desired fairness and objectivity are the reviews and the grades attributed to the paper. The benefits of the proposed solution and emphatic and all reviews were positive with a decent level of confidence (Average grade: 7 Average confidence: 4.00). For this reason, at least a spotlight mention should be attributed to the proposed solution.

**Metareview: Summary, Strengths And Weaknesses:**

I Summary:

- I.1 Investigated Problem:
    - The paper investigates the problem of neural representation for the weights of a pretrained neural network by Proposing NeRN (Neural Representations for Neural Networks)

- I.2 Proposed Solution:
    - Nern performs reconstruction of the parameters of a pretrained convolutional network using predictor MLP and incorporates smoothness constraint over the original network's weights that guides the neural networks towards a better reconstruction.

- I.3 Validity Proof of the Proposed Solution:
    - Authors show the effectiveness of several smoothness-enhancing techniques and validate them on various datasets. Authors refer to two potential and promising applications of their proposed method:
        - weight importance analysis (visualization), where the importance is measured by NeRN’s accuracy;
        - meta-compression, where the predictor is pruned to achieve a disk-size compact representation, possibly without data.

II Strengths

- II.1 From a structural (organization) point of view:
    - The authors did a great job presenting their work. All reviews were unanimous on the quality of the presentation and the structure of the submission

- II.2 From an analytical (development) point of view: Most Reviewers appreciated:
    - The novelty of the method: reconstructing a network’s pretrained weights using a different network was not explored in previous literature, especially when treating the task as a neural representation problem.
    - The clarity of the motivation;
    - The empirical evidence provided to support the case of the proposed solution as the extensive experiments have been conducted:
        - Authors show the effectiveness of several smoothness-enhancing techniques and validate them on various datasets;
    - It is also worth mentioning and the transparency aspect of the submission as open source code is provided for reproducibility purposes

- II.3 From a perspective of soundness (unity, and coherence) and completeness (correctness):
    - The strength points mentioned above are sufficient evidence of the soundness and completeness of the paper.  An additional point reinforcing the strengths mentioned above is the active interaction of the authors during the rebuttal period and their openness to concerns and questions raised by the reviewers.

- III Addressing what can be thought of as weaknesses
    - There isn’t much to mention about weaknesses as:
        - Authors tried to address most of the concerns that reviewers raised and the questions they asked.
        - The presented work is novel of its kind and has a great potential to open up the door for new research in the community
        - Unanimously, the reviewers agree on the acceptance of the submission.

IV. Potential of the paper:

- IV.1 From a Potential perspective (Potential of the paper to the community):
    - The proposed solution has a great potential to be of benefit to the whole community of representation learning. The extension and investigation to other models can also be of great benefit as the presented work provide potential alternatives to hyper-networks and meta-learning.



**Note From Pc:**

if the above contains the word "oral" or "spotlight" please see: "oral" presentation means -> notable-top-5% and "spotlight" means -> notable-top-25%. As stated in our emails, we are disassociating presentation type from AC recommendations

**Summary Of Ac-Reviewer Meeting:**

N/A